# Impact of the COVID-19 lockdown in the United Kingdom on adolescent's time use (CONTRAST study)

**Irina Pokhilenko**[1]*, **Emma Frew**[1], **Marie Murphy**[2], **Miranda Pallan**[2]

**1** Centre for Economics of Obesity, Institute of Applied Health Research, University of Birmingham, Birmingham, United Kingdom, **2** Institute of Applied Health Research, University of Birmingham, Birmingham, United Kingdom

* i.pokhilenko@bham.ac.uk

## Abstract

### Background

The COVID-19 pandemic has led to major changes in everyone's lives, including adolescents. Given that adolescence is a crucial developmental stage, designing strategies to alleviate the impact of the COVID-19 on adolescents is critical. Furthermore, there is a growing literature on the relationship between how adolescents spend their time and impact upon health, nutrition, educational attainment and overall well-being outcomes, and the existence of a socioeconomic gradient with how time is allocated. Therefore, this study explored changes in adolescents' time use during the first COVID-19 lockdown in the UK and the relationship between these changes and individual-level socioeconomic indicators including family affluence, free school meal eligibility, and food insecurity.

### Methods

The data were collected from 11-15-year-olds using an online survey, which contained questions on demographic characteristics, socioeconomic indicators, and time use across a range of activities before and during the first COVID-19 lockdown. Changes in time use in relation to socioeconomic indicators were explored using descriptive and regression analysis.

### Results

687 adolescents completed the survey. There was an overall decrease in the amount of time spent on school work, an increase in screen time, and an increase in sleep duration during the week. Descriptive analysis showed evidence of inequalities with changes in time use. In adjusted regression analyses, family affluence was associated with a greater increase in time spent on socialising with household members and a decrease in time spent on exercise. Free school meal eligibility and experience of food insecurity were associated with an increase in sleep duration.

**Data Availability Statement:** The data associated with this manuscript are available on Zenodo via 10.5281/zenodo.10245892.

**Funding:** This study has received funding from the UKRI Economic and Social Research Council

Impact Acceleration Account programme. The
funders had no role in study design, data collection
and analysis, decision to publish, or preparation of
the manuscript.

**Competing interests:** The authors have declared
that no competing interests exist.

## Discussion

This study contributes to the body of evidence on the impact of the COVID-19 pandemic on
young people and suggests that some of the pre-lockdown inequalities in time allocation
were attenuated as a result of the lockdown. Furthermore, the results underscore the need
for longer term surveillance to monitor changes in time use in adolescents to mitigate the
impact on outcomes over the life course.

## 1. Background

Childhood and adolescence are critical developmental periods. Activities that young people
engage with on a regular basis impact their health, nutrition, cognitive development, educa-
tional achievement and overall well-being. For example, regular involvement in physical activ-
ity is associated with better physical [1, 2] and mental health outcomes [3, 4], and time spent
on learning activities and socialising fosters cognitive skills development [5, 6]. In contrast,
factors such as prolonged screen use and reduced sleep are associated with adverse physical
and psychological effects [4, 7–10], and reduced academic performance [11, 12]. Poor health
and academic outcomes during childhood and adolescence are linked to adverse socioeco-
nomic outcomes over the life course, such as lower employment rates and lower earnings [13–
16], all of which are associated with substantial welfare loss. The continued increase in health
inequalities in the UK has focused attention on how childhood behaviours are linked to the
educational and economic inequalities by household socio-economic status.

One important consideration is how children and adolescents spend time. Evidence sug-
gests that various socioeconomic factors (e.g. parents' education and income) play an impor-
tant role in how young people allocate their time. For example, those from higher income and
higher educated families have been found to spend more time on a wide range of learning and
leisure activities, and less time using screens [5, 17–21] whereas young people from low
income families have been found to spend more time using screens [20–22]. A study by Arnup
and colleagues (2021) explored associations between how children used their time and the
experience of financial hardship and found financial hardship to be associated with increased
screen time, particularly passive screen time (e.g. watching TV), and reduced sleep [23]. This
suggests that children from more affluent families are more likely to allocate time in a manner
that accumulates the human and social capital required for optimal development.

The COVID-19 pandemic and the resulting restrictions had a tremendous impact on every-
one's lives. Due to school closures, children were required to undertake learning activities in
the home environment; other activities such as socialising and organised sports were also sub-
stantially impacted. There is evidence that children and adolescents spent significantly more
time using screens during the pandemic compared to pre-pandemic [24] and research has
shown that COVID-19 had a disproportionate impact on young people from more deprived
backgrounds [25–27].

We expect that the COVID-19 pandemic and the associated restrictions might have exacer-
bated the existing inequalities in how young people allocated their time. Therefore, this study
aimed to explore how adolescents' (age 11–15 years) time use changed during the first
COVID-19 lockdown in the UK introduced in March 2020. Furthermore, the study explored
the relationship between the changes in time use due to the lockdown and family-level socio-
economic indicators including family affluence, free school meal eligibility, and food insecu-
rity as proxy measures of financial hardship. This evidence offers a deeper understanding of

how young people spend their time across different activities, and how the COVID-19 lockdown may have influenced this among families in different socioeconomic positions.

## 2. Methods

### 2.1. Data collection

Data for this study were collected between June 3rd and July 31st 2020 using an online survey. The survey included questions about the participant's characteristics (individual, family, household and school), school learning and time undertaking other activities, eating and physical activity habits, mental wellbeing, sleep, and relationships with family and friends. The survey underwent two rounds of piloting with young people aged 11 to 15 years.

Students enrolled in secondary schools in the UK during the academic year 2019–2020 who were aged between 11 and 15 years were invited to participate. We approached young people to take part in the survey in several ways. We asked schools we already had contact with through other research projects, young people's networks (e.g. GENIUS school food network, Girl Guides and Scouts Associations), and organizations who work with more disadvantaged schools and populations (e.g. Coach Bright, the national network of Enterprise Advisers and The Children's Society) to advertise and distribute the survey. We also advertised the survey on a variety of social media platforms (X (formally known as Twitter), Facebook, Instagram, Snapchat and TikTok). The time-use survey formed part of a larger study on the impact of COVID-19 on learning, eating, physical and other activities; the survey and the study report are available elsewhere [28].

Fig 1 illustrates the timeline of COVID-19 restrictions in England, Scotland and Wales between March and July 2020. The survey was open for completion from 3rd June to 31st July 2020, which coincides with gradual relaxation of the restrictions.

All participants were provided written participant information sheets. Written (online) consent was obtained for all participants. The need for parental/guardian consent was waived by the ethics committee. Part of the consent process was to ask participants to discuss their participation with their parents/guardians before consenting. The ethical approval for this study was sought from The University of Birmingham Research Ethics Committee (ERN_20–0645).

### 2.2. Measurements

**2.2.1. Demographic characteristics.** The following demographic characteristics of respondents were collected: age, gender, ethnicity using UK census classifications, country of residence in the UK (England, Wales, Scotland or Northern Ireland), year group, language spoken at home (English/Welsh or other), and school type (state school, grammar school, private/independent school, special school, pupil referral or behaviour unit, or other).

**2.2.2. Socioeconomic indicators.** Changes in time use were explored in relation to three family-level socioeconomic indicators: family affluence score (FAS), free school meal (FSM) eligibility, and food insecurity, as proxy measures of financial hardship.

FAS was measured using the validated Family Affluence Scale, FAS III [29]. FAS III collects information about a family's material conditions (e.g. ownership of car(s), number of holidays abroad) and constructs a score of family wealth by summing the scores across the indicators. FAS III offers an alternative to family socioeconomic status measures based on income. The FAS III scores can range between 0 (least affluent) and 13 (most affluent). In this study, we first calculated individual FAS scores for each respondent and then grouped the respondents into three groups using tercile cut-offs with 1 being the least affluent and 3 being the most affluent.

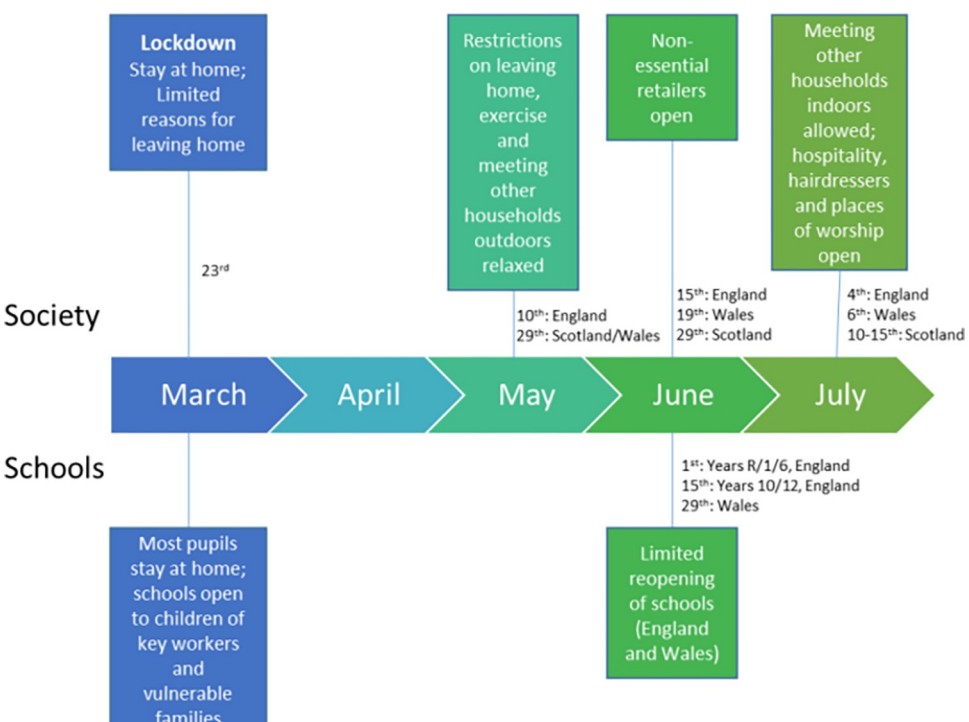

**Fig 1. Timeline of COVID19 restrictions between March and July 2020 in England, Scotland, and Wales [28].**

FSM eligibility was measured by asking the participants if they were eligible for FSMs. In the UK, FSM eligibility is determined by the family's socioeconomic status [30]. Respondents could choose between three options: 'yes', 'no', and 'don't know'.

Two questions taken from a validated scale for measuring food (in)security in older children [31] were used to measure food insecurity:

- 'Has the food that your family bought run out, and you didn't have enough money to get more?'

- 'Have you had to skip a meal because your family didn't have enough money for food?'

The respondents could choose from three answer options: 'a lot', 'sometimes', 'never'. The respondents were divided into two groups: those who experienced food insecurity, defined as responding 'a lot' or 'sometimes' to at least one of these questions, and those who never experienced food insecurity. The measure of financial hardship used in the study by Arnup et al. (2020) [23] contains a similar question asking parents whether they had had to go without meals due to money shortage. Experience of food insecurity as a proxy measure of financial hardship is distinct from income-based measures and tends to be more informative about family living conditions and material deprivation [32].

**2.2.3. Allocation of time.** Several questions on time use before and during the lockdown were asked. The broad categories of time use included were school work undertaken at home, other learning activities outside of school, screen time, exercise, socialising with household members, performing household chores, and sleep. Depending on the expected frequency of involvement in activities, the respondents were asked to indicate the amount of time allocated to each activity either during a typical day or during a typical week. For daily activities, the respondents were asked to indicate the number of hours spent on each activity on a typical day with the choice from: 'no time', 'up to 30 minutes', 'between 30 minutes and 1 hour', '1–3

**Table 1. Measurement of time use.**

| Category of time use | Before lockdown | During lockdown |
|---|---|---|
| School work | Homework set by the school* | Work set by the school (online, by email, on paper or in school)*<br>Live lessons online with my schoolteachers*<br>School work (online or other) set by my parents/other household members* |
| Other learning activities | Activities outside of schoolwork that help me learn new knowledge and skills (e.g. watching a TV documentary, home craft, cooking, learning a musical instrument)* | Same question used |
| | Reading for fun (not homework)* | Same question used |
| | Private academic tuition classes outside school time** | Private academic tuition classes** |
| | Private tuition classes outside school time for music, drama etc.** | Private tuition classes for music, drama etc.** |
| Screen time | Spending time chatting with friends on social media (e.g. Instagram, WhatsApp, Snapchat etc.)* | Same question used |
| | Watching TV/Netflix/YouTube/TikTok etc. (on a TV set, computer, tablet, phone or other device)* | Same question used |
| | Playing games (either yourself or with friends) on a computer/Xbox/PlayStation/phone/ Tablet/other device* | Same question used |
| Exercise | Organised sport/physical activity clubs** | Doing exercise or sport* |
| | Other types of exercise (not formal clubs) e.g. swimming with friends, football in the garden/park etc.** | Exercise or sport** |
| Socialising | Spending time relaxing with other household members* | Same question used |
| Household chores | Helping/doing chores around the house* | Same question used |
| Sleep | Hours of sleep during a typical weekday | Same question used |
| | Hours of sleep during a typical weekend day | Same question used |

*during a typical day
**during a typical week

hours', '4–6 hours', 'more than 6 hours', and 'I don't know'. For weekly activities, the respondents were asked to indicate the frequency of performing each activity with the choice from: 'never', 'less than once per week', 'once per week', '2–3 times per week', '4–5 times per week', and 'more than 5 times per week'. Sleep was measured based on self-reported bed and wake times during weekdays and weekends separately. These questions were adapted from an international survey of children's lives and wellbeing [33] by ensuring the language was natural and familiar and inclusive to all UK-based adolescents. Table 1 provides an overview of the time use categories measured.

## 2.3. Data transformation

The time use data were transformed to enable the analysis of how time use had changed due to the lockdown. For daily activities, a midpoint of the range for each category was taken to estimate changes in time use for each respondent (S1 Table). For example, respondents who reported spending between 0 and 30 minutes per day on a given activity were assigned the value of 0.25, corresponding to 15 minutes or one quarter of an hour.

Additional assumptions were required to estimate the time spent on school work before lockdown. To account for the fact that all children attended school pre-lockdown from 9am–3pm, 5 hours was automatically added to the time spent on homework to give the total time spent on school work pre-lockdown (allowing 1-hour for school break times). To calculate the total time spent on school work during lockdown, the number of hours across the three categories of school work were summed (live lessons, school work set by the school, and school work set by the parents/household members).

To estimate changes in time use on private academic and non-academic tuition, we subtracted the pre-lockdown time use from the lockdown time use and created a binary variable, with 0 indicating no change or increase in frequency, and 1 indicating a decrease in frequency.

Exercise was measured based on the frequency per week. To enable comparison, we assumed a typical exercise session lasted 1 hour. All categories were recoded using range midpoints as indicated in S1 Table.

To calculate the total amount of time spent on exercise before lockdown, we summed the number of hours spent on exercise per week before lockdown. To calculate the difference in time allocated to exercise, the variable indicating the total duration of exercise per week before lockdown was subtracted from the variable indicating the duration of exercise per week during lockdown.

## 2.4. Data analysis

First, the data were analysed descriptively. For the activities measured using the number of hours per day, time allocation for each category and changes in time use (during lockdown vs before lockdown) were described using means and standard deviations for the entire sample, and then separately for the groups defined by the three socioeconomic indicators. For the activities measured based on the frequency per week, the number and percentage of respondents in each category were described for the full sample, and then separately for the groups defined by the socioeconomic indicators.

Second, to compare the changes in the time spent on every activity, paired t-tests were used for the activities measured in the number of hours per day and Wilcoxon signed rank tests were used for the activities based on the frequency per week. Third, we tested the significance of the differences across socioeconomic groups. We conducted unpaired t-tests to test the significance of differences between groups based on food insecurity ('yes' vs 'no') and FSM eligibility ('eligible' vs 'ineligible') variables (for this comparison we excluded participants who did not know their FSM eligibility). To test the significance of differences across the three FAS groups, we conducted one-way ANOVA; where the results were significant, we also ran Tukey post hoc tests for pairwise comparisons to identify which groups were significantly different.

Fourth, we used regression modelling to explore the relationship between changes in each time use category (dependent variable) and each of the three socioeconomic indicators (independent variables) separately, controlling for demographic characteristics (gender, age, ethnicity, language spoken at home, country of residence in the UK, year group, and type of school). Linear and logistic models were used for continuous and binary outcome variables, respectively. All analyses were conducted using Stata version 17.0. P<0.05 was considered statistically significant.

## 3. Results

### 3.1. Descriptive analysis

Demographic and socioeconomic characteristics of the sample are presented in Table 2. A total of 687 people responded to the survey. Approximately half of the sample (49%) had a FAS

**Table 2. Demographic and socioeconomic characteristics of the sample.**

| | N | % or mean (SD) |
|---|---|---|
| Total number of participants | 687 | 100 |
| Gender | 679 | 100 |
| Male | 310 | 45.66 |
| Female | 362 | 53.31 |
| Other/Unknown | 2 | 0.29 |
| Would rather not say | 5 | 0.74 |
| Age (years) | 664 | 13.8 (1.2) |
| Ethnicity | 682 | 100 |
| White | 541 | 79.33 |
| Asian/Asian British | 57 | 8.36 |
| Black / African / Caribbean / Black British | 30 | 4.40 |
| Mixed / Multiple ethnic groups | 36 | 5.28 |
| Other ethnic group | 11 | 1.61 |
| I would rather not say | 7 | 1.03 |
| Year group | 639 | 100 |
| 7 (age 11–12 years) | 183 | 28.64 |
| 8 (age 12–13 years) | 144 | 22.54 |
| 9 (age 13–14 years) | 158 | 24.73 |
| 10 (age 14–15 years) | 154 | 24.10 |
| Country | 661 | 100 |
| England | 624 | 94.40 |
| Scotland | 32 | 4.84 |
| Wales | 5 | 0.76 |
| Main language spoken at home | 686 | 100 |
| English / Welsh | 566 | 82.51 |
| Other | 120 | 17.49 |
| School type | 646 | 100 |
| State | 412 | 63.78 |
| Grammar | 108 | 16.72 |
| Private | 124 | 19.20 |
| Behavioural | 2 | 0.31 |
| Family affluence score groups (equal terciles) | 649 | 100 |
| 1 | 236 | 36.36 |
| 2 | 203 | 31.28 |
| 3 | 210 | 32.36 |
| Eligibility for free school meals | 645 | 100 |
| Yes | 55 | 8.53 |
| No | 542 | 84.03 |
| Don't know | 48 | 7.44 |
| Experience of food insecurity | | |
| Yes | 65 | 10.48 |
| No | 555 | 89.52 |

score of 10 or higher; the majority of the respondents were not eligible for FSMs (84%) and had never experienced food insecurity (90%). S2 Table illustrates the distribution of the respondents across the three socioeconomic indicators.

Furthermore, 476 participants reported which school they went to. Overall, participants represented 147 UK schools, including 77 schools in the West Midlands region (n = 440), 6 schools in Scotland (n = 32), and 3 schools in Wales (n = 4).

### 3.2. Analysis of time use

Table 3 presents the amount of time spent by respondents on daily/weekly activities before and during lockdown. Descriptive analysis shows that time spent on school work during lockdown was approximately two hours less than when usually at school (pre-lockdown). Compared to pre-lockdown, during lockdown there was a statistically significant increase of 1.5 hours in the amount of total screen time with increases across all three categories of screen time, an increase of approximately one hour in the amount of sleep time during weekdays, and a small statistically significant increase in the amount of time spent socialising with household members and doing household chores. There were no significant changes in the time spent on other activities that help young people learn new knowledge and skills such as reading, exercise, and sleep during weekends.

Table 4 illustrates the weekly frequency of private academic tuition and private non-academic tuition before and during lockdown, both of which decreased during lockdown.

### 3.3. Descriptive analysis of the relationship between changes in time use and socioeconomic characteristics

Results of the descriptive analysis of the relationship between changes in time use and socioeconomic characteristics are presented in S3 Table (available in supplementary material). The results of the paired t-tests showed significant decrease in the time spent on school work and significant increase in the screen time and sleep on weekdays in all groups. These results also indicate some evidence of inequalities as a result of the lockdown. For example, adolescents with more disadvantage reported a higher average increase in screen time and a larger decrease in the time spent on school work. On the other hand, adolescents with more advantage (higher

**Table 3. Daily/weekly time use before vs during lockdown (in hours, 1 hour = 1).**

| Activity | Before lockdown Mean (SD) | During lockdown Mean (SD) | Difference of means (during vs pre-lockdown) Mean (SD) |
|---|---|---|---|
| School work (total), including: | 6.58 (1.46) | 4.68 (2.79) | **-1.88 (2.71)**\* |
| *Homework set by the school* | 1.58 (1.46) | n/a | n/a |
| *Work set by the school* | n/a | 3.74 (1.98) | n/a |
| *Live lessons* | n/a | 0.63 (1.16) | n/a |
| *School work set by household members* | n/a | 0.32 (0.91) | n/a |
| Activities outside of schoolwork that help me learn new knowledge and skills | 1.25 (1.44) | 1.21 (1.32) | -0.06 (1.49) |
| Reading for fun | 0.60 (1.04) | 0.64 (1.16) | 0.04 (0.77) |
| Screen time (total), including: | 4.27 (3.74) | 5.78 (4.39) | **1.51 (3.26)**\* |
| *Spending time chatting with friends on social media* | 1.38 (1.55) | 1.85 (1.96) | **0.47 (1.55)**\* |
| *Watching TV/ Netflix/ YouTube/TikTok etc.* | 1.83 (1.69) | 2.44 (2.04) | **0.62 (1.68)**\* |
| *Playing games on a device* | 1.09 (1.64) | 1.48 (1.99) | **0.39 (1.35)**\* |
| Socialising with household members | 1.31 (1.49) | 1.51 (1.52) | **0.17 (1.28)**\* |
| Chores | 0.58 (0.8) | 0.69 (0.96) | **0.08 (0.62)**\* |
| Weekly exercise | 3.68 (2.8) | 3.46 (2.07) | -0.2 |
| Sleep during weekdays | 9.24 (1.09) | 10.18 (1.34) | **0.91 (1.29)**\* |
| Sleep during weekends | 10.35 (1.35) | 10.44 (1.55) | 0.1 (1.27) |

\*significant at $p < 0.05$

**Table 4. Weekly activities before vs during lockdown (weekly frequency).**

| Activity | Response categories | Before lockdown (n (%)) | During lockdown (n (%)) |
|---|---|---|---|
| Private academic tuition | Never | 540 (86.54) | 573 (89.39) |
| | Less than once per week | 15 (2.4) | 10 (1.56) |
| | Once per week | 54 (8.65) | 32 (4.99) |
| | 2–3 times per week | 13 (2.08) | 10 (1.56) |
| | 4–5 times per week | 1 (0.16) | 8 (1.25) |
| | More than 5 times per week | 1 (0.16) | 8 (1.25) |
| Private tuition for music, drama, etc | Never | 418 (66.99) | 475 (74.45) |
| | Less than once per week | 16 (2.56) | 15 (2.35) |
| | Once per week | 120 (19.23) | 99 (15.52) |
| | 2–3 times per week | 58 (9.29) | 40 (6.27) |
| | 4–5 times per week | 10 (1.6) | 5 (0.78) |
| | More than 5 times per week | 2 (0.32) | 4 (0.63) |

FAS and FSM ineligible) reported significant decreases in the time spent on exercise, whereas no significant changes were observed in more disadvantaged adolescents. We also observed a higher number of adolescents with disadvantage reporting a decrease in the frequency of academic and non-academic tuition (S4 Table).

The differences in changes in time use between different groups were only significant for the time spent on school work, sleep, and exercise (S3 and S5 Tables). Adolescents with more disadvantage had a significantly larger decrease in the time spent on school work (applicable to lower FAS and FSM eligibility); i.e. adolescents in the least affluent FAS group spent on average 2.4 hours less on school work per day compared to 1.7 and 1.5 hours decrease in FAS groups 2 and 3 respectively. We also observed larger increase in the sleeping time on weekdays (applicable to all three indicators) and weekends (applicable only to food insecurity) among the adolescents with more disadvantage. For example, adolescents in FAS group 1 slept on average 1.1 hours longer on weekdays during the lockdown, while adolescents in FAS groups 2 and 3 had a slightly lower increase of 0.9 and 0.8 hours per day on average, respectively. Furthermore, we observed significant differences in the time spent on exercise across FAS groups; adolescents in FAS groups 2 and 3 had decreases in the time spent on exercise, -0.6 and -0.4 hours per week on average respectively, compared to adolescents in FAS group 1, who had a non-significant increase of 0.3 hours per week on average (S3 and S5 Tables).

## 3.4. Regression analysis of the relationship between changes in time use and socioeconomic characteristic

Table 5 presents the results of the regression analyses exploring the association between changes in time use and the socioeconomic indicators, controlling for respondents' demographic characteristics (full regression analyses presented in S6 Table). Separate models have been used to explore each of the three socioeconomic indicators as explanatory variables.

FAS was found to be significantly associated with changes in the time spent on exercise and socialising with household members. The average change in the time spent on exercise in adolescents in the 2nd FAS group was significantly more negative, compared to adolescents in the 1st (lowest scoring) FAS group, which is in line with the descriptive analysis. Thereby, both analyses indicate that the pre-lockdown socioeconomic inequalities in the time spent on exercise were somewhat attenuated (S3 Table). Adolescents in the 3rd FAS group (highest scoring) had a greater increase in the time spent on socialising with household members. FSM eligibility and the experience of food insecurity were only found to be significantly associated with

**Table 5. Resuts of the regression analysis of changes in time use.**

| Time use category | Socioeconomic indicators | | | | |
|---|---|---|---|---|---|
| | Model 1** | | Model 2** | | Model 3** |
| | FAS group (1 –reference category) | | Free school meal eligibility (Not eligible–reference category) | | Experience of food insecurity (No– reference category) |
| | 2 (β (95% CI)) | 3 (β (95% CI)) | Eligible (β (95% CI)) | Don't know (β (95% CI)) | Yes (β (95% CI)) |
| School work (total) | 0.30 (-0.20–0.80) | -0.03 (-0.54–0.48) | -0.61 (-1.31–0.08) | -0.09 (-0.85–0.67) | -0.33 (-1.00–0.34) |
| Activities outside of schoolwork that help me learn new knowledge and skills | -0.17 (-0.48–0.15) | 0.19 (-0.13–0.51) | 0.03 (-0.42–0.49) | 0.29 (-0.19–0.76) | -0.05 (-0.46–0.36) |
| Reading for fun | 0.02 (-0.15–0.18) | 0.15 (-0.02–0.31) | -0.11 (-0.34–0.12) | **0.36* (0.12–0.60)** | 0.01 (-0.20–0.22) |
| Screen time (total) | 0.28 (-0.42–0.97) | 0.36 (-0.34–1.06) | 0.45 (-0.57–1.48) | -0.48 (-1.50–0.55) | 0.64 (-0.27–1.54) |
| Spending time chatting with friends on social media | 0.00 (-0.32–0.32) | -0.03 (-0.35–0.30) | 0.33 (-0.14–0.80) | -0.03 (-0.50–0.44) | 0.20 (-0.22–0.62) |
| Watching TV/ Netflix/ YouTube/TikTok etc. | 0.22 (-0.14–0.57) | 0.22 (-0.14–0.58) | -0.26 (-0.76–0.24) | -0.19 (-0.71–0.33) | 0.16 (-0.30–0.62) |
| Playing games on a device | 0.02 (-0.26–0.30) | 0.14 (-0.15–0.42) | 0.37 (-0.03–0.76) | -0.24 (-0.66–0.17) | 0.17 (-0.19–0.53) |
| Socialising with household members | 0.14 (-0.13–0.42) | **0.34* (0.06–0.62)** | -0.22 (-0.61–0.17) | -0.09 (-0.50–0.32) | -0.17 (-0.53–0.18) |
| Chores | -0.03 (-0.16–0.10) | -0.03 (-0.16–0.10) | 0.08 (-0.11–0.27) | -0.16 (-0.36–0.03) | 0.03 (-0.13–0.18) |
| Exercise | **-0.80* (-1.35 - -0.26)** | -0.53 (-1.08–0.01) | -0.57 (-1.35–0.21) | 0.07 (-0.72–0.87) | 0.05 (-0.64–0.75) |
| Sleep during weekdays | -0.22 (-0.49–0.05) | -0.16 (-0.44–0.12) | **0.40* (0.01–0.79)** | **0.76* (0.36–1.16)** | 0.30 (-0.05–0.65) |
| Sleep during weekends | 0.16 (-0.11–0.44) | 0.23 (-0.05–0.52) | 0.19 (-0.21–0.58) | **0.75* (0.35–1.16)** | **0.44* (0.09–0.80)** |
| Academic tuition | 1.56 (0.66–3.67) | 1.88 (0.82–4.30) | 0.18 (0.02–1.36) | 1.13 (0.38–3.41) | 0.73 (0.24–2.20) |
| Non-academic tuition | 0.95 (0.52–1.74) | 0.94 (0.51–1.72) | 0.64 (0.24–1.70) | 0.78 (0.31–1.94) | 0.86 (0.39–1.92) |

*significant at p<0.05; Family Affluence Score (FAS), confidence interval (CI)

**each regression model also included gender, age, ethnicity, language spoken at home, country of residence in the UK, year group, and type of school as covariates

changes in sleep duration, which is also in line with the results of the descriptive analysis. FSM eligible pupils had a higher increase in sleep duration during weekdays compared to FSM ineligible pupils, whereas the experience of food insecurity was only found to be significantly associated with a higher increase in the weekend sleep duration.

Some of the demographic characteristics also explained the associations tested in the regression analysis (S6 Table). Compared to boys, girls had a significantly higher increase in the time spent watching TV and a larger decrease in the time spent playing online games. Older adolescents had a higher increase in the time spent chatting with friends on social media, doing chores, exercising, and sleeping. Compared to White adolescents, Asian/Asian British adolescents had a higher increase in the sleep duration and Black/ African/Caribbean/ Black British adolescents had a higher increase in the time spent reading. Finally, school type was also significantly associated with some of the changes in time use. For example, private school attendance was positively associated with increased time spent on school work, using social media to chat to friends, and the frequency of non-academic tuition.

## 4. Discussion

The aim of this study was to examine how young people changed their allocation of time in relation to selected family-level socioeconomic indicators during the first UK COVID-19 lockdown in 2020. Overall, there was a significant decrease in the amount of time spent on school work, an increase in screen time including the time spent on social media, watching television, and playing games, and increase in sleep time during weekdays. The descriptive analysis showed some evidence of socioeconomic inequalities in time allocation during lockdown. For example, before the lockdown there were negligible differences among the respondents in the time they spent on school work (potentially due to set school times all adolescents have to adhere to), but during the lockdown pupils from more disadvantaged groups reported spending significantly less time on school work compared to the pupils from more socioeconomically advantaged groups. The latter may be associated with limited access to and the need to share technology (e.t. laptops and tablets) in more deprived households. However, there were decreases in time spent on exercise in the groups with higher socioeconomic advantage compared to small non-significant increases in less advantaged groups, and higher increases in weekday sleep duration in those with lower socioeconomic advantage. For some other categories of time use, we found that pre-lockdown inequalities persisted during the lockdown, for example overall screen time was higher across more disadvantaged groups pre-lockdown, but screen time increased consistently across all socioeconomic groups during the lockdown.

When other covariates were controlled for within the regression models, the relationship between time use and socioeconomic variables was attenuated, potentially due to the fact that some of the covariates were better able to explain the distribution in the changes in time use. For example, private school attendance was significantly associated with the changes in several categories of time use, including the time spent on school work, exercise, non-academic tuition, and using social media to chat to friends. This suggests that private school attendance may be considered to be a socioeconomic variable that could better explain these associations. Only associations between measures of socioeconomic status and changes in the time spent on exercise, socialising with household members, and weekday sleep duration remained when controlling for other characteristics, suggesting the attenuation of pre-lockdown socioeconomic inequalities during the COVID-19 lockdown.

A decrease in the time spent on school work is consistent with other published studies [34], and can be explained by school closures and adaptation to the new format of learning during the lockdown. Some early projections predict significant losses in earnings for the students whose schooling was affected by the pandemic [25, 35]. It remains to be seen what the long-term impact of COVID-19 and associated lost schooling will be [36]. The average screen time in our sample went up from 4.3 to 5.8 hours a day—an increase consistent with the published evidence [24, 37]. With no universally accepted screen time threshold recommendations for adolescents it is difficult to put this result into a policy context. Some evidence suggests that high levels of smartphone and social media use may be harmful [38, 39]. However, there has been a debate about the potential harms and benefits from using screens, especially in light of the changes brought about by the pandemic and the resulting shift to more online learning [40–42].

We also found an overall small decrease in the time spent on exercise, with significant decrease in more affluent groups compared to slight non-significant increase in less affluent groups. At the same time, adolescents with more advantage had higher levels of exercise pre- and during the lockdown compared to the adolescents is less affluent groups. Larger decreases in this group could be attributed to the restrictions placed on organized sports and exercise during the lockdown, which more affluent adolescents may engage with more due to the costs

associated with these types of activities, among other reasons. Furthermore, more affluent adolescents reported a significantly larger increase in the time spent socialising with household members; given that pre-lockdown they reported spending less time on this activity compared to adolescents in less affluent groups, this indicates a narrowing of a gap between the groups. These changes could be explained by the nature of jobs in more affluent households, with parents being more likely to switch to remote working and thereby be able to spend more time with their children.

In accordance with other studies [43], we observed an increase in sleep duration during the week across all respondents, with the most significant increase among adolescents who were either eligible for FSMs or did not know their FSM eligibility status. In the UK, teenagers are recommended to sleep at least 9 hours per day [44] and the majority in our sample were achieving this pre-lockdown, with the exception of the latter two groups (pupils eligible for FSMs and pupils who did not know if they were eligible) who reported sleep duration slightly lower than 9 hours. The higher increase in these two groups during lockdown helped attenuate the pre-lockdown differences, which might have led to additional health and well-being benefits for these respondents. Although it is important to note that we did not consider neither sleep quality nor sleep efficiency in our study.

We also observed several differences for the change in time allocation by gender. Girls were found to have a higher increase in the time spent watching television/YouTube, etc, while boys were found to spend more time playing videogames during the lockdown. This might reflect pre-existing preferences by gender in relation to how adolescents spend their time using screens. For example, there is evidence that boys tend to spend more time playing computer games compared to girls, while girls tend to spend more time on social media [45, 46]. In our sample, there was a higher increase in the time spent on social media among girls compared to boys, but this was not statistically significant. Statistically significant increases in the time spent watching television among girls could be explained by the fact that during the lockdown, when activity options were restricted, girls simply preferred watching tv to playing videogames. It is important to note that time allocation differences by socioeconomic indicators were our main focus, and we did not explore time use before and during lockdown in relation to the demographic characteristics of the respondents.

In our analysis, we considered three socioeconomic indicators (FAS, FSM eligibility, and experience of food insecurity); however, little is known about how well these indicators correlate with each other. The results of the descriptive analysis indicate that there is an overlap between different groups, but that there are also some discrepancies. For example, a number of respondents with the highest FAS score reported having had an experience of food insecurity. This might explain why we did not observe consistent trends in changes in time use in relation to these indicators with the exception of school time. FAS was at times contradicting FSM eligibility and food insecurity; for example, trends in changes in the screen time differed depending on the selected socioeconomic indicator. We posit that FAS might have been less informative of the respondents' socioeconomic status compared to FSM eligibility and food insecurity because of the way we used this indicator in our analysis, i.e. using a statistical method to define groups, which resulted in the middle and upper groups both having relatively high FAS scores.

## Methodological reflections

From a methodological perspective, this study has several strengths. First, the data were collected in June to July 2020, relatively soon after the initial COVID-19 lockdown measures were eased. The respondents therefore were assumed to have a good recollection of their time use

during the lockdown. Second, we collected a wide range of data on participants' demographic and socioeconomic characteristics and their time use across a wide range of typical daily activities, which gave us a comprehensive picture of how the respondents spent their time. The use of multiple socioeconomic indicators in our study may also be considered a strength, as they were not equivalent, their inclusion provided more nuanced insights into changes in time use.

The study also had some limitations. First, given the target group for this study as well as the ongoing pandemic, the recruitment strategy was heavily reliant on social media and this could have led to the overrepresentation of respondents from certain population subgroups, e.g. those who have access to smartphones. This may explain our sample composition–the majority of the participants came from a more affluent background. FSM eligibility in our sample was at 8.5%, while nationally it was 17.3% in the academic year 2019/2020 [47]. It is important to note though that participants' FSM status was self-reported and 7.5% of the respondents did not know their FSM status, which might partially explain the difference with the national level FSM eligibility. 10.5% of our respondents reported having the experience of food insecurity; this is similar to the estimate of 9.4% for July 2022 [48]. However in our study, the participants were asked whether they had ever experienced food insecurity, while in the report of the Food Foundation, the respondents were asked about their experience of food insecurity in the past month. Furthermore, there was a higher proportion of students attending private schools in our sample (19.2%) compared to the general population in England (6.4%).

Second, this survey collected self-reported data which might have been subject to self-report bias. Third, it was necessary to make a number of assumptions to conduct our analysis, for example, with the time use data, we used midpoint values for each category (i.e. 2 hours for the category 1–3 hours). This might have led to the loss of granularity in our results. Fourth, the data for this survey were collected in 2020 and reflected changes in time allocation during the first COVID-19 lockdown in the UK. Given that the acute nature of the pandemic lasted until 2022 that led to multiple lockdowns in the UK of which the full impact is still to be gauged, we cannot state with certainty that the effects seen in our study are applicable to these later lockdowns nor to countries other than the UK.

## Implications for policy

In economics, time is considered to be a scarce resource that has an economic value. Depending on how time is allocated by an individual, it can be used as an input in the production of goods such as health and social capital, which are considered valuable by society [49, 50]. Therefore, policies targeting people's time allocation can result in economic benefits, particularly for adolescents, as they form habits and establish educational trajectories at this age. For example, Fiorini & Keane (2014) suggest that 'a reallocation of children's time that favours educational activities by substituting away from less productive ones would have a positive effect on cognitive skill' [5]. Given that cognitive skills are associated with lifetime earnings [51], influencing adolescents' time allocation decisions and their investment in formative activities should be encouraged [52]. Del Boca et al (2017) also suggest that adolescents cultivate the sense of agency over their own development, therefore, such policies should target adolescents themselves rather than parents [52].

Our results also add to the body of evidence on socioeconomic disparities. In line with other studies, in the descriptive analysis we observed a disproportionate impact on adolescents with lower family socioeconomic positioning, although our adjusted analysis suggested other factors may explain this observed relationship. Furthermore, the use of multiple socioeconomic measures in our study was instrumental in revealing some underlying dynamics, such

as the experience of food insecurity by more affluent groups, that can support the ongoing debate about expanding the FSM eligibility criteria in England [53, 54].

## Implications for further research

The results of this study also have implications for further research. First, in light of the short-term nature of this study, a follow-up exploration of how time allocation among adolescents has changed throughout the course of the pandemic and beyond can provide useful evidence to inform what policies are needed to mitigate its impact. Some studies suggest that behavioural changes, such as decreased exercise and increased screen time, persisted after the first COVID-19 lockdown was lifted [55]. Considering the detrimental health impacts of these changes (e.g. increases in childhood obesity levels), further research of long-term time allocation trends is warranted.

Second, given the variation in our results in relation to the selected socioeconomic indicators, our study indicates that inclusion of multiple measures of SES may be beneficial in exploratory research to detect nuances in the relationships with time use. However, further research is needed to understand how different measures relate to each other and which socioeconomic dimensions are most informative of young people's involvement in specific activities and health and educational outcomes. It is also important to consider the limitations of each of these measures. While FAS III is a validated instrument, it is less sensitive to more affluent respondents [29]. FSM eligibility is a commonly used socioeconomic indicator in the UK, but it is not transferable to other countries, and even within the four UK nations there are differences in the FSM eligibility requirements [56]. As for food insecurity, we used two questions from the validated scale for measuring food security in older children [31]. However, there is no consensus on what constitutes food insecurity and what the most appropriate measure is [57], and so there is a need for development of a robust food insecurity measure that is suitable for young people. Also, our findings suggest that private school attendance might be considered as another socioeconomic indicator as it was significantly associated with the changes in several categories of time use.

Third, there have been several recent studies that investigated time allocation [23, 58] and they used different approaches to measuring time. In the study by Arnup et al. (2021), the researchers asked parents to fill in time use diaries for their children, which was considered an appropriate method given the young age of target population [23]. Huls et al. (2022) measured time allocation among adults by asking them to indicate the exact number of hours spent on each activity [58]. Given the limitations associated with how time was measured in our study and the fact that other researchers reported potential bias and measurement error due to their selection of measurement method, further methodological research is needed to explore which method of measuring time use is most appropriate.

## 5. Conclusion

This study investigated the impact of the first COVID-19 lockdown in the UK on how adolescents allocated their time on daily activities in relation to their socioeconomic characteristics. Our results suggest a significant decrease in the amount of time spent on school work, increase in screen time, and an increase in sleep time during the working week. Socioeconomic indicators were only associated with the changes in weekday sleep duration and the time spent on exercise and socialising with household members, when other factors were adjusted for, suggesting the attenuation of pre-lockdown inequalities in time allocation. Overall, we suggest longer term surveillance to monitor how time-use varies within and between adolescent populations given the impact on outcomes over the life course.

## Supporting information

**S1 Table. Recoding of time use variables.**
(DOCX)

**S2 Table. Socioeconomic characteristics of the sample.**
(DOCX)

**S3 Table. Descriptive changes in time use by socioeconomic indicators (in hours, 1 hour = 1).**
(DOCX)

**S4 Table. Changes in the frequency of academic and non-academic tuition and results of Wilcoxon signed rank test (z-scores).**
(DOCX)

**S5 Table. Results of the Tukey post hoc tests testing the significance of time use changes across family affluence score groups.**
(DOCX)

**S6 Table. Results of the regression analysis of changes in time use.**
(DOCX)

## Author Contributions

**Conceptualization:** Irina Pokhilenko, Emma Frew, Marie Murphy, Miranda Pallan.

**Data curation:** Marie Murphy, Miranda Pallan.

**Formal analysis:** Irina Pokhilenko, Emma Frew, Marie Murphy, Miranda Pallan.

**Funding acquisition:** Emma Frew, Marie Murphy, Miranda Pallan.

**Investigation:** Irina Pokhilenko, Emma Frew, Marie Murphy, Miranda Pallan.

**Methodology:** Irina Pokhilenko, Emma Frew, Marie Murphy, Miranda Pallan.

**Project administration:** Irina Pokhilenko, Emma Frew, Marie Murphy, Miranda Pallan.

**Resources:** Miranda Pallan.

**Software:** Irina Pokhilenko.

**Supervision:** Emma Frew, Miranda Pallan.

**Validation:** Miranda Pallan.

**Writing – original draft:** Irina Pokhilenko.

**Writing – review & editing:** Irina Pokhilenko, Emma Frew, Marie Murphy, Miranda Pallan.

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
