## [Decision Letter · Decision Letter 0]

3 May 2024

PONE-D-23-40253Impact of the COVID-19 lockdown in the United Kingdom on adolescent’s time use (CONTRAST study)PLOS ONE

Dear Dr. Pokhilenko,

Thank you for submitting your manuscript to PLOS ONE. After careful consideration, we feel that it has merit but does not fully meet PLOS ONE’s publication criteria as it currently stands. Therefore, we invite you to submit a revised version of the manuscript that addresses the points raised during the review process.

We look forward to receiving your revised manuscript.

Kind regards,

Marta Tremolada, Ph.D.

Academic Editor

PLOS ONE

 [This study has received funding from the UKRI Economic and Social Research Council Impact Acceleration Account programme.].  

3. We noted in your submission details that a portion of your manuscript may have been presented or published elsewhere. [The study described in this manuscript was a part of a larger study, the CONTRAST study. The CONTRAST study report was published on the CONTRAST Study website presenting the overall descriptive results. We refer to this report in the manuscript. Please also see: https://www.birmingham.ac.uk/documents/college-mds/applied-health/contrast-study/contrast-report-06-04-2021-accessible.pdf. Our study uses a part of the CONTRAST study dataset for an in-depth exploration of changes in adolescents’ time use during the COVID-19 lockdown using regression analysis. This is not discussed in the study report. Therefore, we do not consider this submission to constitute dual publication.] Please clarify whether this publication was peer-reviewed and formally published. If this work was previously peer-reviewed and published, in the cover letter please provide the reason that this work does not constitute dual publication and should be included in the current manuscript.

Additional Editor Comments:

Dear authors of "Impact of the COVID-19 lockdown in the United Kingdom on adolescent’s time use (CONTRAST study)", I think that the paper should be valid and interesting after reviewing it along their suggestions.

The topic is new, the scientific soundness of the paper is present, but there are some limits that should be overcome with a precise and clear work on the text.

After revisions, the paper will be reconsidered for publication.

Reviewers' comments:

Reviewer's Responses to Questions

**Comments to the Author**

1. Is the manuscript technically sound, and do the data support the conclusions?

Reviewer #1: Yes

Reviewer #2: Partly

2. Has the statistical analysis been performed appropriately and rigorously? 

Reviewer #1: Yes

Reviewer #2: Yes

3. Have the authors made all data underlying the findings in their manuscript fully available?

Reviewer #1: Yes

Reviewer #2: Yes

4. Is the manuscript presented in an intelligible fashion and written in standard English?

Reviewer #1: Yes

Reviewer #2: Yes

5. Review Comments to the Author

Reviewer #1: Thank you, for the opportunity to review this study who addresses an important area of inquiry regarding the impact of the pandemic on adolescents, particularly in relation to changes in time use and socioeconomic disparities.

The background provided is comprehensive, effectively framing the research within the context of the COVID-19 pandemic and the unique challenges it poses to adolescent development. The rationale for investigating the relationship between time use and socioeconomic indicators is well justified.

The methodology section provides clear details regarding data collection and analysis methods, utilizing an online survey to gather information from 11-15-year-olds about their demographic characteristics, socioeconomic status, and time use patterns. The inclusion of both descriptive and regression analyses adds depth to the investigation, allowing for a comprehensive exploration of the research questions.

Regarding the questionnaire, how were the questions selected?

Was the questionnaire validated?

The results section presents key findings regarding changes in adolescent time use during the COVID-19 lockdown.

The discussion section effectively contextualizes the findings within the broader literature on the pandemic's impact on young people and emphasizes the need for continued surveillance to monitor changes in time use and mitigate potential long-term consequences. However, additional discussion on the limitations of the study, such as potential biases inherent in online surveys and the generalizability of findings beyond the UK context, would strengthen the paper.

Overall, the study provides valuable insights about the adolescent time use during Covid-19 pandemic. With some minor revisions and further consideration of limitations, this research has the potential to make a significant impact in both academic and practical contexts.

The authors should correct the references, as they didn’t used the journal reference style.

Reviewer #2: The author group have presented some interesting data about time use in adolescents during lockdown. there are some novel findings reported although there are also some significant limitations that can be tracked through teh attached document. In summary the data collected are biased in that there is a significant overrepresentation of affluent children. for example double the proportion of children who attend private school. there are also limitations in how the data are reported and the depth of analysis, reporting and interpretation. Some of the stongest sections are in the limitations and further research parts of the paper. Time use allocation changes during lockdown re very important to gain deeper insight into and these studies are important as we reflect back over the impact of COVID-19 on the development of younger generations

6. PLOS authors have the option to publish the peer review history of their article (what does this mean?). If published, this will include your full peer review and any attached files.

Reviewer #1: No

Reviewer #2: No

---

## [Author Response · Author response to Decision Letter 0]

14 Jun 2024

Editor’s comments

Response: Thank you for referring us to PLOS ONE’s style requirements. We carefully reviewed them and revised the format of the manuscript accordingly. We revised author information, and submitted supporting information separately as individual files in line with the instructions. We also made sure to use the PLOS reference output style in EndNote as one of the reviewers made a note about the format of the references.

 [This study has received funding from the UKRI Economic and Social Research Council Impact Acceleration Account programme.]. 

Response: We corrected the Role of Funder statement as indicated below and included this information in the cover letter. 

‘This study has received funding from the UKRI Economic and Social Research Council Impact Acceleration Account programme. The funders had no role in study design, data collection and analysis, decision to publish, or preparation of the manuscript.’

3. We noted in your submission details that a portion of your manuscript may have been presented or published elsewhere. [The study described in this manuscript was a part of a larger study, the CONTRAST study. The CONTRAST study report was published on the CONTRAST Study website presenting the overall descriptive results. We refer to this report in the manuscript. Please also see: https://www.birmingham.ac.uk/documents/college-mds/applied-health/contrast-study/contrast-report-06-04-2021-accessible.pdf. Our study uses a part of the CONTRAST study dataset for an in-depth exploration of changes in adolescents’ time use during the COVID-19 lockdown using regression analysis. This is not discussed in the study report. Therefore, we do not consider this submission to constitute dual publication.] Please clarify whether this publication was peer-reviewed and formally published. If this work was previously peer-reviewed and published, in the cover letter please provide the reason that this work does not constitute dual publication and should be included in the current manuscript.

Response: We thank the editor for this comment. The report we refer to was neither peer-reviewed nor formally published in an academic journal. The report contains the results of the descriptive analysis of the data collected in the CONTRAST study. More specifically, it contains descriptive analysis of time use data by family affluence score, but it does not report on any further quantitative analysis. Whereas the manuscript under review with PLOS ONE describes detailed quantitative analysis of time use data using regression accounting for demographic and socioeconomic characteristics of participants.

Response: We copied the abstract from the manuscript into the online submission system.

Additional Editor Comments:

Dear authors of "Impact of the COVID-19 lockdown in the United Kingdom on adolescent’s time use (CONTRAST study)", I think that the paper should be valid and interesting after reviewing it along their suggestions.

The topic is new, the scientific soundness of the paper is present, but there are some limits that should be overcome with a precise and clear work on the text.

After revisions, the paper will be reconsidered for publication.

Response: We thank the editor for acknowledging the merit of our work and for providing support in revising the format and the content of the manuscript. We attach the revised version of the manuscript with and without tracked changes, as well as the rebuttal letter, in which we respond to each of the reviewers’ comments. We hope that the revised version of the paper will now be acceptable for publication in PLOS ONE.

Reviewers' comments:

Reviewer's Responses to Questions

Comments to the Author

1. Is the manuscript technically sound, and do the data support the conclusions?

Reviewer #1: Yes

Reviewer #2: Partly

2. Has the statistical analysis been performed appropriately and rigorously? 

Reviewer #1: Yes

Reviewer #2: Yes

3. Have the authors made all data underlying the findings in their manuscript fully available?

Reviewer #1: Yes

Reviewer #2: Yes

4. Is the manuscript presented in an intelligible fashion and written in standard English?

Reviewer #1: Yes

Reviewer #2: Yes

5. Review Comments to the Author

Reviewer #1: Thank you, for the opportunity to review this study who addresses an important area of inquiry regarding the impact of the pandemic on adolescents, particularly in relation to changes in time use and socioeconomic disparities.

The background provided is comprehensive, effectively framing the research within the context of the COVID-19 pandemic and the unique challenges it poses to adolescent development. The rationale for investigating the relationship between time use and socioeconomic indicators is well justified.

The methodology section provides clear details regarding data collection and analysis methods, utilizing an online survey to gather information from 11-15-year-olds about their demographic characteristics, socioeconomic status, and time use patterns. The inclusion of both descriptive and regression analyses adds depth to the investigation, allowing for a comprehensive exploration of the research questions.

Regarding the questionnaire, how were the questions selected?

Was the questionnaire validated?

The results section presents key findings regarding changes in adolescent time use during the COVID-19 lockdown.

The discussion section effectively contextualizes the findings within the broader literature on the pandemic's impact on young people and emphasizes the need for continued surveillance to monitor changes in time use and mitigate potential long-term consequences. However, additional discussion on the limitations of the study, such as potential biases inherent in online surveys and the generalizability of findings beyond the UK context, would strengthen the paper.

Overall, the study provides valuable insights about the adolescent time use during Covid-19 pandemic. With some minor revisions and further consideration of limitations, this research has the potential to make a significant impact in both academic and practical contexts.

The authors should correct the references, as they didn’t used the journal reference style.

Reviewer #2: The author group have presented some interesting data about time use in adolescents during lockdown. there are some novel findings reported although there are also some significant limitations that can be tracked through teh attached document. In summary the data collected are biased in that there is a significant overrepresentation of affluent children. for example double the proportion of children who attend private school. there are also limitations in how the data are reported and the depth of analysis, reporting and interpretation. Some of the stongest sections are in the limitations and further research parts of the paper. Time use allocation changes during lockdown re very important to gain deeper insight into and these studies are important as we reflect back over the impact of COVID-19 on the development of younger generations

6. PLOS authors have the option to publish the peer review history of their article (what does this mean?). If published, this will include your full peer review and any attached files.

Do you want your identity to be public for this peer review? For information about this choice, including consent withdrawal, please see our Privacy Policy.

Reviewer #1: No

Reviewer #2: No

Responses to reviewers’ comments

Reviewer #1: Thank you, for the opportunity to review this study who addresses an important area of inquiry regarding the impact of the pandemic on adolescents, particularly in relation to changes in time use and socioeconomic disparities.

The background provided is comprehensive, effectively framing the research within the context of the COVID-19 pandemic and the unique challenges it poses to adolescent development. The rationale for investigating the relationship between time use and socioeconomic indicators is well justified.

The methodology section provides clear details regarding data collection and analysis methods, utilizing an online survey to gather information from 11-15-year-olds about their demographic characteristics, socioeconomic status, and time use patterns. The inclusion of both descriptive and regression analyses adds depth to the investigation, allowing for a comprehensive exploration of the research questions.

Regarding the questionnaire, how were the questions selected?

Was the questionnaire validated?

The results section presents key findings regarding changes in adolescent time use during the COVID-19 lockdown.

The discussion section effectively contextualizes the findings within the broader literature on the pandemic's impact on young people and emphasizes the need for continued surveillance to monitor changes in time use and mitigate potential long-term consequences. However, additional discussion on the limitations of the study, such as potential biases inherent in online surveys and the generalizability of findings beyond the UK context, would strengthen the paper.

Overall, the study provides valuable insights about the adolescent time use during Covid-19 pandemic. With some minor revisions and further consideration of limitations, this research has the potential to make a significant impact in both academic and practical contexts.

The authors should correct the references, as they didn’t used the journal reference style.

Response: We would like to thank Reviewer 1 for their time and for the feedback on the manuscript. We also note the positive feedback on the relevance, potential impact, and technical aspects of our work. Below, we respond to Reviewer 1’s individual comments.

Reviewer 1’s comment: Regarding the questionnaire, how were the questions selected? Was the questionnaire validated?

Response: We thank the reviewer for this comment. The questionnaire included questions on socio-demographic characteristics. We selected validated measures (e.g. Family Affluence Scale) to measure participants’ socio-economic status. Time-use questions were adapted from an international survey of children’s lives and wellbeing by ensuring the language was natural and familiar and inclusive to all UK-based adolescents. We added this information and the reference to the report to the manuscript as follows:

‘These questions were adapted from an international survey of children’s lives and wellbeing [33] by ensuring the language was natural and familiar and inclusive to all UK-based adolescents.’ (p. 4, 147-149)

The questionnaire underwent two rounds of piloting with young people aged 11-15. We’ve also added this information to the manuscript:

‘The survey underwent two rounds of piloting with young people aged 11 to 15 years.’ (p. 2, 80-81)

Reviewer 1’s comment: However, additional discussion on the limitations of the study, such as potential biases inherent in online surveys and the generalizability of findings beyond the UK context, would strengthen the paper.

Response: We thank the reviewer for this comment. We have now listed several limitations of our recruitment strategy including the overrepresentation of more affluent participants and biases associated with self-reported data. We have also added reflections about the applicability of the findings outside of the UK as follows:

‘Given that the acute nature of the pandemic lasted until 2022 that led to multiple lockdowns in the UK of which the full impact is still to be gauged, we cannot state with certainty that the effects seen in our study are applicable to these later lockdowns nor to countries other than the UK.’ (p. 17, 391-393)

Reviewer 1’s comment: The authors should correct the references, as they didn’t used the journal reference style.

Response: We thank the reviewer for this comment. We reviewed the format of the references to ensure that we follow PLOS ONE guidelines.

Reviewer #2: The author group have presented some interesting data about time use in adolescents during lockdown. there are some novel findings reported although there are also some significant limitations that can be tracked through teh attached document. In summary the data collected are biased in that there is a significant overrepresentation of affluent children. for example double the proportion of children who attend private school. there are also limitations in how the data are reported and the depth of analysis, reporting and interpretation. Some of the stongest sections are in the limitations and further research parts of the paper. Time use allocation changes during lockdown re very important to gain deeper insight into and these studies are important as we reflect back over the impact of COVID-19 on the development of younger generations

Response: We would like to thank Reviewer 2 for their time and for the feedback on the manuscript. We note the positive feedback on the relevance and potential impact of our work. We also acknowledge that our work is not without limitations, including overrepresentation of young people with a higher socioeconomic status in our sample, and we reflect on this extensively in the discussion section. Furthermore, due to the richness of the data, the depth of the analysis and reporting had to be condensed to fit within one manuscript. We respond to f

---

## [Decision Letter · Decision Letter 1]

3 Sep 2024

Impact of the COVID-19 lockdown in the United Kingdom on adolescent’s time use (CONTRAST study)

PONE-D-23-40253R1

Dear Dr. Pokhilenko,

We’re pleased to inform you that your manuscript has been judged scientifically suitable for publication and will be formally accepted for publication once it meets all outstanding technical requirements.

Kind regards,

Marta Tremolada, Ph.D.

Academic Editor

PLOS ONE

Additional Editor Comments (optional):

I saw all your revisions and I think that now the paper could be publishable in this current form.

Reviewers' comments:

Reviewer's Responses to Questions

**Comments to the Author**

1. If the authors have adequately addressed your comments raised in a previous round of review and you feel that this manuscript is now acceptable for publication, you may indicate that here to bypass the “Comments to the Author” section, enter your conflict of interest statement in the “Confidential to Editor” section, and submit your "Accept" recommendation.

Reviewer #1: All comments have been addressed

2. Is the manuscript technically sound, and do the data support the conclusions?

Reviewer #1: Yes

3. Has the statistical analysis been performed appropriately and rigorously? 

Reviewer #1: Yes

4. Have the authors made all data underlying the findings in their manuscript fully available?

Reviewer #1: Yes

5. Is the manuscript presented in an intelligible fashion and written in standard English?

Reviewer #1: Yes

6. Review Comments to the Author

Reviewer #1: In my opinion the authors have answered to all the reviewers comments and the paper can be publish in the present form.

7. PLOS authors have the option to publish the peer review history of their article (what does this mean?). If published, this will include your full peer review and any attached files.

Reviewer #1: No

---

## [Editor Report · Acceptance letter]

24 Sep 2024

PONE-D-23-40253R1 

PLOS ONE

Dear Dr. Pokhilenko, 

I'm pleased to inform you that your manuscript has been deemed suitable for publication in PLOS ONE. Congratulations! Your manuscript is now being handed over to our production team.

Kind regards, 

on behalf of

Dr. Marta Tremolada 

Academic Editor

PLOS ONE